# Influence of Chromophoric Electron-Donating Groups on Photoinduced Solid-to-Liquid Transitions of Azopolymers

**DOI:** 10.3390/polym12040901

**Published:** 2020-04-13

**Authors:** Jian Xu, Bin Niu, Song Guo, Xiaolei Zhao, Xiaoli Li, Jinwen Peng, Weixing Deng, Si Wu, Yuanli Liu

**Affiliations:** 1Guangxi Key Laboratory of Optical and Electronic Materials and Devices, College of Materials Science and Engineering, Guilin University of Technology, Guilin 541004, China; xujian8814@163.com (J.X.); niubingzz@163.com (B.N.); ahguosong@163.com (S.G.); z1842655946@163.com (X.Z.); lixiaoliqwq@163.com (X.L.);; 2CAS Key Laboratory of Soft Matter Chemistry, Department of Polymer Science and Engineering, University of Science and Technology of China, Jinzhai Road 96, Hefei 230026, China; siwu@ustc.edu.cn; 3Nanning Proberoo Energy Materials Co., Ltd. 41 Jianye Road, Liangqing District, Nanning 530221, China

**Keywords:** azopolymer, electron-donating groups, photoisomerization, solid–liquid transitions

## Abstract

The photoinduced solid-to-liquid transitions property of azobenzene-containing polymers (azopolymers) enables azopolymers with various promising applications. However, a general lack of knowledge regarding the influence of structure of the azobenzene derivatives on the photoinduced liquefaction hinders the design of novel azopolymers. In the present study, a series of azopolymers with side chains containing azobenzene unit bearing alkyl electron-donating groups were synthesized. The photoisomerization and photoinduced liquefaction properties of newly synthesized azopolymers were investigated. Alkyl-based electron-donating group significantly facilitate the photoisomerization process of azopolymers in solution, as the electron-donating ability of substituents increased, the time required for photoisomerization of azopolymers continually deceased. Meanwhile, the electron-donating group can drastically accelerate photoinduced solid-to-liquid transitions of azopolymers, the liquefaction rate of obtained azopolymers gradually getting quicker as the electron-donating ability of substituents increased. This study clearly demonstrates that the electron-donating group that bearing in the azobenzene group of polymer side chain play an essential role on the photoinduced solid-to-liquid transitions of azopolymers, and hence, gives an insight into how to design novel azopolymers for practical applications.

## 1. Introduction

Polymers containing azobenzene chromophores (azopolymers) have been extensively investigated in past few decades for their potential applications [1,2,3]. Azobenzene and its derivatives can undergo reversible trans–cis isomerization upon irradiation with UV or visible light at certain wavelength [4,5,6]. Induced by the reversible change of molecular shape of azobenzene moiety, azopolymers demonstrate a variety of photoresponsive properties, and result in a number of practical applications, such as optical information storage [7,8], actuators [9,10], surface gratings [11,12,13], light-driven molecular switches [14,15], sustained-release drugs [16,17,18], light-controlled microfluidics and other photonic devices [19,20,21,22].

Photoinduced solid-to-liquid transitions (liquefaction) property of azopolymers have attracted worldwide attention since it makes polymers more easily to reprocessable, reshapable and healable [23]. Usually, reprocessing of solid thermoplastic polymers requires liquefying the polymers by heating them above a specific temperature and solidify upon cooling down below that temperature. As a contrast, photoinduced liquefaction can occur at room temperature, some solid azopolymers can “flow” upon light irradiation. These unique properties make photoinduced liquefaction suitable for reprocessing of polymer, switchable adhesives, high-resolution lithography and printing in a manner of light-driven and remote control. Dr. Si Wu’s team successfully introduced azobenzene group into the side chain of polyacrylic acid polymer, the glass transition temperature (TG) of the polymer was controlled reversibly by alternating ultraviolet (365 nm)/visible (530 nm), and realized the reversibly photoinduced solid–liquid transitions [24]. Dr. Yoshida’s team reported that azopolymers capable of undergoing photoinduced liquefaction could be used as reworkable adhesives to enable on-demand bonding and debonding by irradiation with visible and UV light [25].

Macroscopic photoresponsive properties of azopolymers are ascribed to photoisomerization of azobenzene units and subsequent chain motions of polymer [26,27]. In general, the motions of azopolymer can be categorized into three levels [28,29]. The chromophore motion at the molecular level is the first level, which is accompanied often with change in the dipole moment and with change in the orientation of the chromophore. The motion at the domain level is the second level, which requests the binding of the chromophore to the polymer matrix to form an ordered structure, phase transitions and amplified motion at the domain level are often found in Langmuir−Blodgett films and liquid crystals. The third level is the macroscopic motion, which is the massive response of azopolymer materials to external light stimuli, e.g., the visible change in pattern of films or change in the shape of elastomers and gels. In recent years there has been an increasing interest in the investigation of the effect of fundamental molecular structure on photoinduced macroscopic motion of azopolymer. Dr. Si Wu’s team reported the effect of different lengths of spacers between azopolymer backbone and azobenzene unit on the photoresponsive behavior [30]. Although extensive efforts have been devoted to developing azopolymers capable of undergoing photoinduced liquefactions, there are several critical issues that need to be addressed in order to further elucidate the correlation between the structure and photoresponsive property of azopolymers.

Herein, a series of azopolymers with side chain containing azobenzene units with alkyl-based electron-donating substituents were synthesized by atom transfer radical polymerization (ATRP). We investigated the effect of alkyl-based electron-donating groups on the photophysical properties of prepared azopolymers. The structure of the monomers was characterized by nuclear magnetic resonance, the molecular weight of the polymers and its distribution were determined by gel permeation chromatography, the photoisomerization process of these polymers in solution was analyzed by UV-vis spectrophotometry and the photoinduced liquefaction was recorded with optical microscope after UV irradiation. The results show that the alkyl-based electron-donating group can accelerate the photoisomerization process of azopolymers in solution, and the electron-donating group can drastically facilitate photoinduced solid–liquid transitions of azopolymers as well. This research indicates that electron-donating groups that bearing in the azobenzene group of polymers play an essential role on the photoinduced solid-to-liquid transitions of azopolymers, and hence, gives an insight into how to design novel azopolymers by tuning the structure of attached azobenzene unit for practical applications.

## 2. Materials and Methods

### 2.1. Materials

Aniline, p-methylaniline, 4-ethylaniline, 4-isopropylaniline and 4-tert-butylaniline were purchased from Shanghai Adamas Reagent Co., Ltd. (Shanghai, China). Phenol, 6-chloro-1-hexanol provided by Shanghai Adamas reagents Co., Ltd. (Shanghai, China). Triethylamine, acryloyl chloride and sodium nitrite are also supplied by Adamas Reagent, Ltd. (Shanghai, China). 2-Bromoisobutyryl bromide and N,N,N',N,'N''-pentamethyldiethylenetriamine (PMDETA) were purchased from Sigma-Aldrich Trading Co., Ltd. (Shanghai, China). Hydrochloric acid (HCl), glacial acetic acid (CH_3_COOH), sodium hydroxide (NaOH), potassium carbonate (K_2_CO_3_) and potassium iodide (KI) were purchased from Xilong Science Co., Ltd. (Shantou, China). The purified water used in all the experiments was prepared with a water purification system AXLM1820 (Asura Technology Development Co., Ltd., Chongqing, China).

### 2.2. Characterization

^1^H NMR spectra of samples were performed on a 400 MHz NMR instrument (AVANCE III HD 400 MHz, Swiss Bruker, Brooke, Switzerland) and tetramethylsilane was used as an internal standard in deuterated chloroform (CDCl_3_). The ultraviolet-visible (UV-vis) spectra were recorded on a PerkinElmer Lambda 365 spectrometer (PerkinElmer, Waltham, MA, USA). The molecular weight and molecular weight distribution were conducted on an Agilent 1260 HPLC system (Agilent Technologies Inc., Santa Clara, CA, USA), tetrahydrofuran (THF) was used as eluent at 20 °C at a flow rate of 0.5 mL/min while using a refractive index (RI) detector and polystyrene calibration. The photo-liquefied photos were acquired by the microscope (XP-300C) (Shanghai Caikon Optical Instrument Co., Ltd. Shanghai, China). The ultraviolet light source is provided by Mightex (Mightex, Toronto, Ont., Canada).

### 2.3. Synthesis of Monomer

The overall synthesis procedures of monomers M-n-Azo (n = 1–5) were shown in Figure 1. The synthesis route for each individual monomer are shown in Appendix A. Taking M-1-Azo as an example, in a 100 mL three-neck flask equipped with a magnetic stirrer. 4-hydroxyazobenzene (1.98 g, 10 mmol) was dissolved in 20 mL of N,N-dimethylformamide (DMF), 3.45 g of K_2_CO_3_ and 50 mg of KI was then added, followed with addition of 6-bromo-1-hexanol (2.17 g, 12 mmol) dropwisely, and the reaction was carried out at 110 °C for 12 h. After cooling, 200 mL of deionized water was added to obtain a yellow precipitate. After extraction, the crude product was purified with a silica gel column (petroleum ether: ethyl acetate = 3:1) to give compound 5 (R = H). In a 250 mL three-neck flask equipped with a magnetic stirrer, 2.98 g of compound 5 (R = H) was dissolved in 100 mL of THF, and triethylamine (3 g, 30 mmol) was added and stirred. Acryloyl chloride (2.7 g, 30 mmol) was added and the reaction was carried out at 0 °C for 2 h. Filter and then remove the solvent, column chromatography (petroleum ether: ethyl acetate = 10:1) gave an orange solid of monomer M-1-Azo.

### 2.4. Synthesis of Azopolymers

The synthesis procedures of azopolymers P-n-Azo (*n* = 1–5) were shown in the last step of Figure 1. All five azopolymers were prepared by the same method. Taking P-1-Azo as an example, M-1-Azo (352 mg, 1 mmol) was dissolved in 2 mL anisole, followed by the addition of CuBr (15 mg, 0.1 mmol), 2-bromoisobutyryl bromide (EBIB; 5.8 mg, 25 μmol) and PMDETA (35 mg, 0.2 mmol). The solution was frozen in liquid nitrogen, evacuated for 20 min using a vacuum pump, and then thawed under a nitrogen atmosphere. This cycle was repeated three times. Subsequently, the solution was reacted at 80 °C for 12 h. A basic alumina column was used to remove copper from the reaction solution, following which the solution was precipitated three times to give a yellow powder. The other monomers were polymerized in a similar manner.

## 3. Results and Discussion

### 3.1. Nuclear Magnetic Spectrum of Monomers

Figure 2 shows the ^1^H NMR spectra (Swiss Bruker, Brooke, Switzerland) of monomers M-n-Azo (*n* = 1–5). The chemical shift of protons in each compound was assigned in detail in the support information. A characteristic chemical shift of alkyl groups in the bellow spectra was observed, which clearly demonstrated that alkyl based electronic donating groups, methyl group, ethyl group, isopropyl group and tert-butyl group were successfully bared on the para position of azobenzene moieties.

### 3.2. Characterization of Azopolymers

Molecular weight and molecular weight distribution of all azopolymers are measured and shown in Table 1.

### 3.3. Photoisomerization Properties of Azopolymers

Previous study has demonstrated that the introduction of a soft segment with proper length between the azo unit and the polymer backbone could increase the movable space of the azobenzene unit and consequently facilitate the photoisomerization process [30]. The azopolymers investigated in this study all contained six methylene groups between the azobenzene unit and the polyacrylate backbone. The photoisomerization behavior of azopolymers was investigated by using UV−vis spectroscopy. All P-n-Azo polymers showed distinct trans-to-cis photoisomerization in ethyl acetate before and after UV irradiation (Figure 3a–e). Before irradiation, P-n-Azo polymers exhibit a typical strong π−π* absorption band in the UV range and a weak n−π* absorption band in the visible light range. Upon UV-light irradiation (365 nm, 29 mW cm^−2^), all azopolymers decreased the π−π* band and increased the n−π* band, thus demonstrating trans-to-cis isomerization of P-n-Azo, which is well known for azo chromophores.

Interestingly, the structure of the azobenzene unit significantly affected the photoisomerization behaviors of obtained azopolymers, as demonstrated in Figure 3. The azopolymer P-1-Azo shows the slowest photoisomerization rate, it takes 120 s to complete the isomerization upon UV irradiation (Figure 3a). For azopolymer P-2-Azo, containing azobenzene with a methyl group in the para position as electron-donating group, shows a much quicker photoisomerization rate, it took only 70 s to complete the isomerization process (Figure 3b). Moreover, as the electron-donating ability of substituents increased, the time required for photoisomerization of azopolymers continually deceased. For P-3-Azo and P-4-Azo, containing azobenzene with an ethyl group and isopropyl group in the para position as the electron-donating group, it took 60 and 50 s to complete the isomerization process, respectively (Figure 3c,d). For P-5-Azo, containing a tert-butyl group as electron-donating substituents in the azobenzene moieties, it showed the fastest photoisomerization rate, it took 40 s to complete the isomerization process (Figure 3e). These results clearly demonstrated that the photoisomerization behavior of azopolymers was closely related with the chromophore structure and electron-donating capacity of the substitutes.

### 3.4. Photoinduced Solid-to-Liquid Properties of Azopolymers

The trans−cis photoisomerization behaviors of azopolymers can consequently induce the macroscopic motion and phase change of polymeric materials [29]. Figure 4 showed the photoinduced solid-to-liquid transition of the series of synthesized azopolymers. When P-1-Azo powders were illuminated with 365 nm ultraviolet light (Figure 4a). The irregularly shaped P-1-Azo solid powders changed into liquified drops. Adjacent liquified drops fused into single drops. This observation indicates that ultraviolet illumination induced flow of the azopolymer. In addition, powders of the rest four azopolymers with electron-donating substituents in the azobenzene moieties also underwent a liquefied process upon ultraviolet illumination (Figure 4b–e), suggesting that all of the synthesized azopolymers underwent a photoinduced solid-to-liquid transition upon ultraviolet illumination.

Similarly, the structure of azobenzene unit significantly affected the photoinduced solid-to-liquid transition properties of obtained azopolymers, as demonstrated in Figure 4a. The azopolymer P-1-Azo, did not contain the electron-donating group, showed the slowest liquefaction rate and it took 30 min to complete the solid-to-liquid transition process upon UV irradiation. In contrast, those azopolymers with electron-donating substituents in the azobenzene moieties shows a relative faster liquefaction rate. For azopolymer P-2-Azo, which contains a methyl group in the para position of azobenzene as electron-donating group, it took only 16 min to complete the liquefaction process (Figure 4b). Furthermore, as the electron-donating ability of substituents increased, the time costed for liquefaction of azopolymers continually deceased. For P-3-Azo and P-4-Azo, containing azobenzene with an ethyl group and isopropyl group in the para position as the electron-donating group, it took 15 and 12 min to complete the liquefaction process, respectively (Figure 4c,d). For P-5-Azo, containing a tert-butyl group as electron-donating substituents in the azobenzene moieties, it showed the fastest photo-induced solid-to-liquid transition when irradiated with UV light and it took 10 min to complete the liquefaction process (Figure 4e). These results clearly demonstrated that the trend of photoinduced solid-to-liquid transition of azopolymers was determined by the chromophore structure and electron-donating capacity of the substitutes and totally in agreement with the photoisomerization process.

## 4. Conclusions

In the present study, in order to understand the structure–property relationship and reveal the influence of chromophoric electron-donating groups on photoinduced solid-to-liquid transition properties, a series of azopolymers P-n-Azo with side chains containing azobenzene unit bearing alkyl-based electron-donating groups, methyl, ethyl, isopropyl and tert-butyl groups were synthesized. The photoisomerization and photoinduced solid-to-liquid properties of newly synthesized azopolymers were investigated. The electron-donating ability of alkyl groups significantly affected the photoisomerization process of azopolymers in solution, as the electron-donating ability increased, the time required for photoisomerization of azopolymers continually deceased. Similarly, the electron-donating ability of alkyl groups remarkably affected the photoinduced solid-to-liquid transitions of azopolymers as well. The liquefaction rate of azopolymers drastically increased as the electron-donating ability of substituents increased. This work clearly demonstrated that the electron-donating group in the azobenzene group of the polymer side chain played an essential role on the photoinduced solid-to-liquid transitions of azopolymers, and hence, gives insight into how to design novel azopolymers based on the structure–property relationship for practical applications.

## Figures and Tables

**Figure 1 polymers-12-00901-f001:**
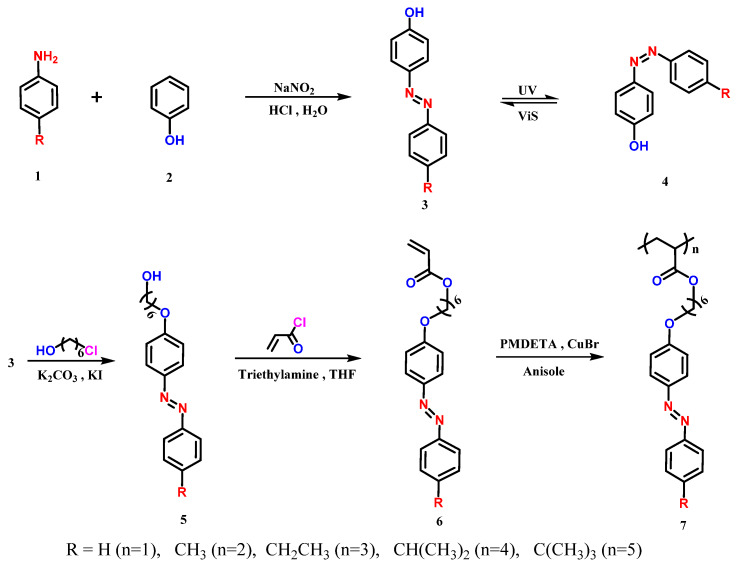
Schematically illustration of the synthesis of monomers M-n-Azo (6) and azopolymers P-n-Azo (7), *n* = 1–5.

**Figure 2 polymers-12-00901-f002:**
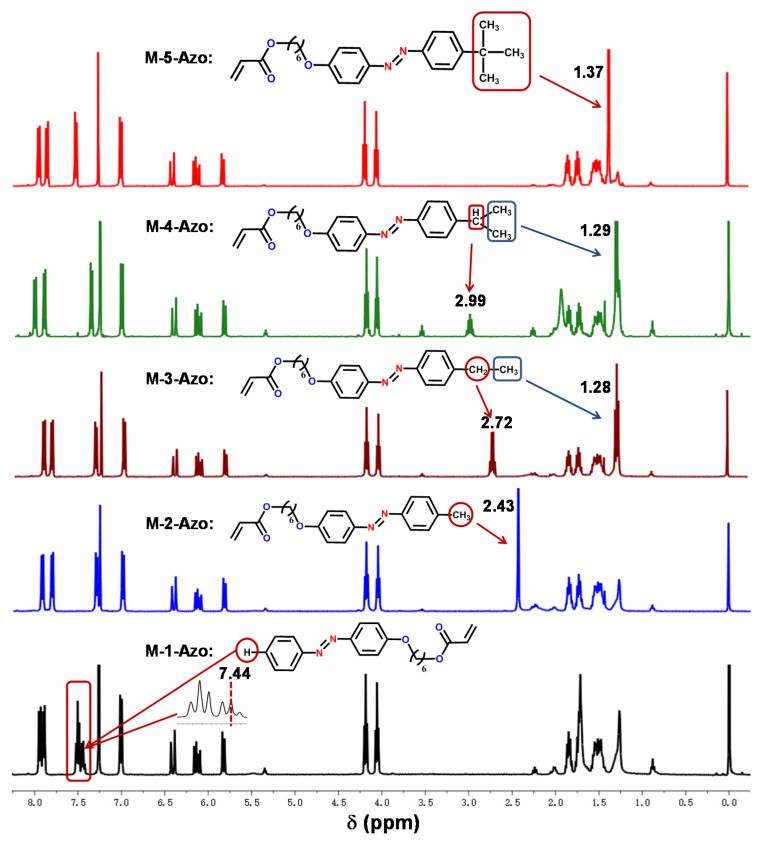
^1^H NMR of monomers M-n-Azo (*n* = 1–5).

**Figure 3 polymers-12-00901-f003:**
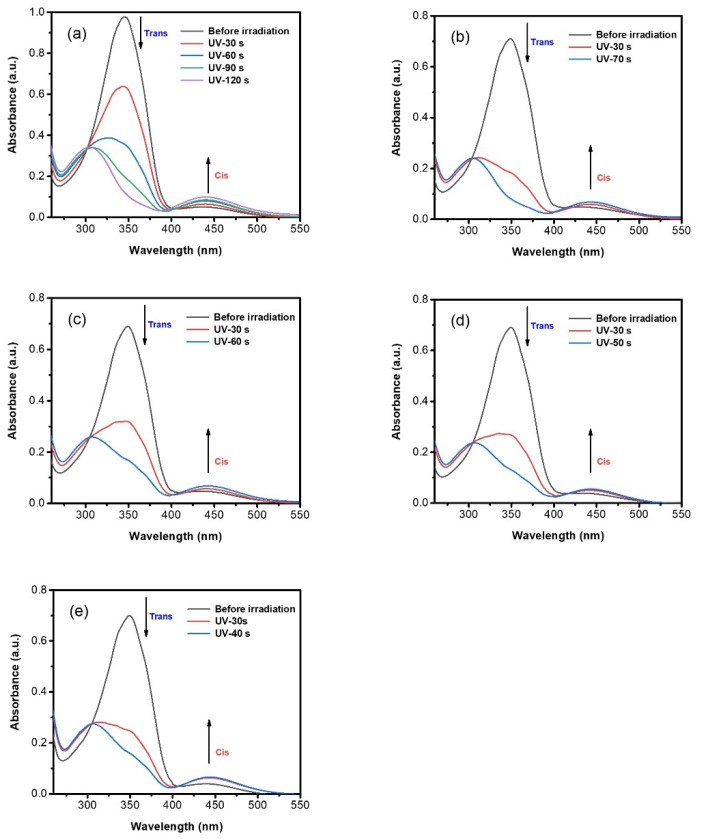
Absorption spectra of azopolymers P-n-Azo (n = 1~5, (**a**–**e**)) after irradiation with 365 nm UV (5 mW/cm^2^) light in ethyl acetate.

**Figure 4 polymers-12-00901-f004:**
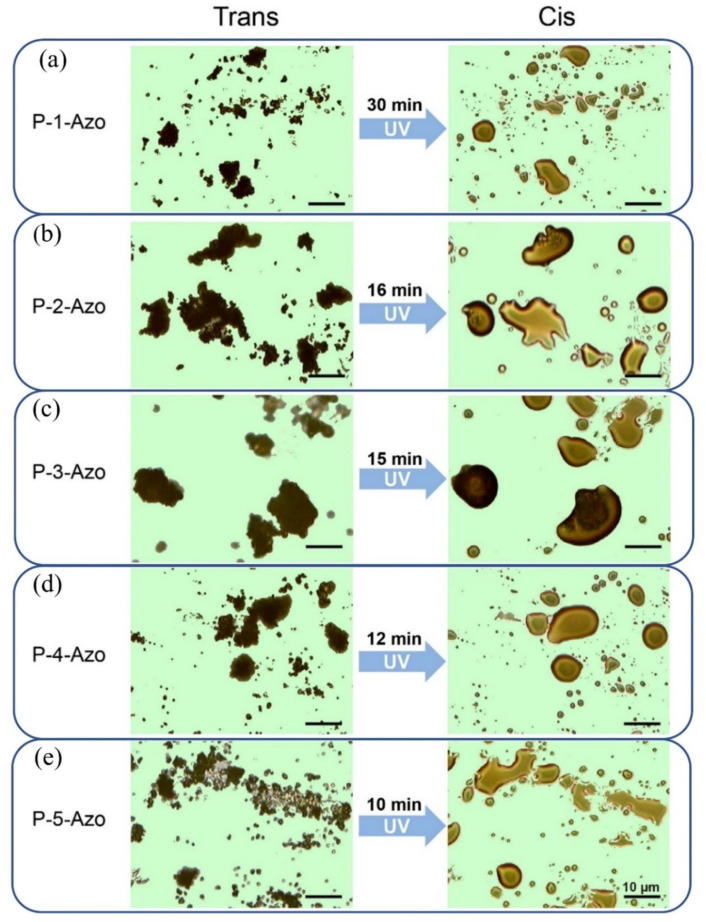
Photoinduced solid-to-liquid transition of azopolymers P-n-Azo (*n* = 1–5, (**a**–**e**)) upon UV (125 mW/cm^2^) irradiation.

**Table 1 polymers-12-00901-t001:** Molecular weight and molecular weight distribution of azopolymers.

Azopolymers ^a)^	P-1-Azo	P-2-Azo	P-3-Azo	P-4-Azo	P-5-Azo
***M*_n_ (g/mol)**	4.2 × 10^3^	6.1 × 10^3^	4.4 × 10^3^	5.8 × 10^3^	6.2 × 10^3^
***M*_w_ (g/mol)**	5.4 × 10^3^	8.5 × 10^3^	5.7 × 10^3^	7.2 × 10^3^	7.7 × 10^3^
***M*_z_ (g/mol)**	7.2 × 10^3^	11 × 10^3^	7.0 × 10^3^	8.8 × 10^3^	9.2 × 10^3^
***PDI***	1.29	1.39	1.31	1.24	1.24

^a)^*M*_n_, number-average molecular weight; *M*_w_, weight-average molecular weight; *M*_z_, Z-average molecular weight; *PDI*, polydispersity index.

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
