# Peer review of "Influence of Chromophoric Electron-Donating Groups on Photoinduced Solid-to-Liquid Transitions of Azopolymers"

_polymers, 2020, doi:10.3390/polym12040901_

Round 1

Reviewer 1 Report

The Authors investigated the photoisomeration and photoinduced solid-to-liquid properties of synthesized azopolymers. The work demonstrated that the electron-donating group that bearing in the azobenzene group of polymer side chain play an essential role on the photoinduced S2L transition.

The manuscript can be accepted for publication in Polymers after minor revision:

  • in chapter 2.2 Author wrote that they used GPC for analysis of polymers. However, the details of analysis are in Chapter 3.3. I think that all these information, e.g. eluent, flow rate etc. should be placed in Chapter 2.2 Characterization.
  • Figure 4: Authors in manuscript write Fig.4a, Figure 4b-e. However I don't see any letters on the Figure 4. Also, the scale is not visible, it would be better to change the color of the scale to black.
  • Figure 4: the scale is 10 nm (nanometers) on the right bottom picture. I think that is mistake. Authors should place a correct value (micrometers?) The information about method of making these pictures should be placed in Methods Chapter (it was made with stereoscope, microscope?)
  • There are some minor mistyping, e.g. line 203 and 221 where the new sentence starts with small letter. Authors should check the manuscript carefully once again in order to avoid such misprints.

Reviewer 2 Report

The paper reports on a systematic study of the influence of electron-donating substituents on the solid to liquid transition of azobenzene containing polymers.

The work is well designed and performed and can be accepted in its present form.

Author Response

Dear Reviewer

        We appreciate you for taking your precious time to review our manuscript and give us chance to publish our paper. 

       best regards

       on behalf all of authors